# Experience of Family Caregivers in Long-Term Care Hospitals During the Early Stages of COVID-19: A Phenomenological Analysis

**DOI:** 10.3390/healthcare12222254

**Published:** 2024-11-12

**Authors:** Hye-Ji Cha, Mi-Kyeong Jeon

**Affiliations:** Department of Nursing, Changwon National University, Changwon 51140, Republic of Korea; chaheyjee@naver.com

**Keywords:** caregiver, COVID-19, pandemics, long-term care, hospitals, phenomenology

## Abstract

Background/Objectives: This study investigated the experiences of inpatient family caregivers who experienced restrictions in meeting their family members owing to the changed visitation guidelines of long-term care hospitals during the early stages of the coronavirus disease 2019 (COVID-19). We aimed to deepen our understanding of the nature of these caregivers’ experiences. Methods: The participants were family caregivers of patients in long-term care hospitals during the early stages of the COVID-19 pandemic. We collected data from nine inpatient caregivers from April to July 2021. Individual interviews and data were analyzed using Colaizzi’s phenomenological method. Results: Participants’ experiences were classified into the following three themes: (1) a prison-like long-term care hospital bound by strict COVID-19 prevention rules; (2) growing affection for unreachable parents; and (3) adaptation to a new, safer daily life. Participants had difficulty communicating with their families living under quarantine conditions during COVID-19 due to non-face-to-face situations. However, amid the long-term COVID-19 situation, participants overcame these challenges through efforts to facilitate communication. Conclusions: The results of this study can be used as basic data to support the smooth communication between patients and caregivers in long-term care hospitals in the event of an infectious disease outbreak in the future, to alleviate emotional stress, and to minimize the weakening of families.

## 1. Introduction

The coronavirus disease 2019 (COVID-19) is a respiratory illness caused by SARS-CoV-2 (severe acute respiratory syndrome coronavirus-2). It originated in Wuhan, China, in December 2019 and quickly spread across China and globally. On 11 March 2020, the World Health Organization declared it a pandemic [1]. South Korea, also hit by the global spread of COVID-19, eventually reclassified it as a Category 4 infectious disease on 31 August 2023. Later, on 1 May 2024, Korea lowered the national infectious disease crisis level from the previous “alert” status to that of “interest.” Nevertheless, even in 2024, wearing masks is still recommended in nursing hospitals in Korea as a preventive measure against respiratory viruses such as COVID-19. In nursing hospitals where older adult patients are mainly present, personal quarantine rules such as wearing masks and disinfecting hands are recommended because of their high vulnerability to infection. In addition, hospitals targeting older adults, who comprise a high-risk group, still maintain certain quarantine measures and, even inside hospitals, wearing masks is actively recommended in crowded environments [2].

To prevent the spread of COVID-19—which is transmitted through respiratory droplets—the South Korean government and the Korea Disease Control and Prevention Agency initially recommended canceling or postponing large gatherings, events, and performances. The government implemented social distancing and everyday quarantine measures, leading to significant changes in daily life. These changes had effects on mental health, with 48% of the South Korean population experiencing depression and stress levels exceeding those seen during the MERS (Middle East Respiratory Syndrome) outbreak, the Gyeongju/Pohang earthquakes, and the Sewol ferry disaster [3]. Similarly, in the US, 20–33% of the population reported experiencing depression and anxiety during this period [4,5]. Thus, concerns related to COVID-19 led to increased depression, anxiety, fatigue, stress, and negative emotions [6].

Although all age groups are susceptible to COVID-19, older adults with weakened immune systems and underlying health conditions have been identified as particularly vulnerable. Consequently, older adult care facilities and long-term care hospitals across Korea were advised to follow stringent infection control guidelines. Patients and staff in nursing care facilities, long-term care institutions, and psychiatric hospitals, among other facilities, were subject to stricter guidelines. For example, long-term care hospitals conducted mandatory tests for admission and restricted family visits to prevent the spread of infection. Such restrictions on family visits caused stress for patients owing to the isolation from their families [7]. Family members, meanwhile, experienced guilt, anxiety, and stress from being unable to visit their loved ones or participate in their care [8,9]. A closer look at the specific impact of the COVID-19 pandemic on long-term care hospitals reveals a global surge in deaths and infections in such facilities. For example, in some countries, between 40% and 80% of all COVID-19-related deaths involved long-term care residents. In particular, Canada experienced a larger-than-expected increase in deaths in long-term bcare hospitals during the first two waves of the pandemic, with COVID-19 having a profound emotional and physical impact on both caregivers and patients [10,11]. Given that COVID-19 had been ongoing for over a year at the time of this study and was recognized as a long-term issue rather than one that could be resolved quickly, it likely placed a heavy burden on the families of patients in long-term care hospitals. Moreover, the potential for future infectious diseases could continue to generate anxiety and concern in both patients and their families.

Patients admitted to long-term care hospitals tend to have severe conditions such as advanced dementia, cancer, or stroke [12]. With the rapid aging of South Korea’s population, long-term care hospital admission rates are rising, and the families of these patients are becoming increasingly fatigued by long-term caregiving and family conflicts. Families caring for older adult patients who require long-term care experience psychological and financial burdens, isolation, and loneliness [13]. Additionally, the patients’ sense of isolation has since increased due to visit restrictions and non-face-to-face management, leading to worsening depression and anxiety. Research results have also shown that their caregivers also experienced stress, emotional exhaustion, and guilt due to not being able to visit [11]. There is a close correlation between the caregivers’ emotional health and the need for long-term care. Caregivers have even been called “hidden patients” because they, too, require care, attention, and support [14,15,16], but they often receive less attention compared with the patients themselves [17,18].

The COVID-19 pandemic caused families and patients to be physically and emotionally separated, which has led to serious emotional problems such as stress, fatigue, guilt, and isolation. However, previous studies on the COVID-19 pandemic have mainly focused on the fatigue experiences of nurses and medical staff [19,20,21]. During the COVID-19 pandemic, hospitalization in long-term care hospitals not only affected patients but also placed considerable psychological and financial burden on their families. This emphasizes the need to understand the experiences of these family caregivers. However, such research is lacking; particularly, research on the difficulties experienced by families and patients in relation to non-face-to-face communication is limited.

This study used a phenomenological approach to better understand family caregivers’ experiences with long-term care hospital patients during the COVID-19 pandemic. By examining these caregivers’ subjective experiences and perceptions, we aimed to gain a deeper understanding of the essence of their experiences. Specifically, we aimed to answer the question, “What is the essence of the experiences of family caregivers of long-term care hospital patients during the COVID-19 pandemic?” By exploring family caregivers’ subjective experiences and insights, we gained a deeper understanding of their lived experiences.

## 2. Materials and Methods

### 2.1. Research Design

This study explored family caregivers’ experiences with the long-term care of hospital patients during the COVID-19 pandemic. We conducted an inductive study using Colaizzi’s phenomenological method [22] to understand the essence of the participants’ experiences. The phenomenological method is suitable for identifying and describing the vivid experiences of participants from their perspective and elucidating the essence and structure of those experiences [22].

### 2.2. Selection of Study Participants

The participants were the family caregivers of long-term care patients in hospitals in City C during the COVID-19 pandemic. In Colaizzi’s phenomenological method, the legitimacy of the sample size can be determined based on the saturation of data and the in-depth exploration of experiences. In qualitative research, saturation is important, and once reached, no new information comes out; thus, no additional participants are required. Generally, the sample size is determined based on the research topic and the participants’ experiences; hence, 9–10 participants are sufficient [23]. The participants were selected through introductions by the nursing department heads of each long-term care hospital. The introduction by the nursing heads was random and did not influence the participants’ decision to participate. During the study, we expanded the number of participants using snowball sampling, in which the current participants introduced other potential participants. Specific inclusion and exclusion criteria are listed in Table 1.

Consent was obtained from all participants, who were assured that the hospital staff would not be aware of their participation decisions. Participants expressed no concerns regarding coercion or negative consequences tied to their involvement in the study. Table 2 and Figure 1 present the general characteristics of the participants.

### 2.3. Data Collection

We collected data through in-depth individual interviews from 23 April 2021 to 9 July 2021. Before collecting the data, we developed primary and supplementary interview questions based on prior research [8,24,25,26] on caregivers’ experiences in long-term care hospitals and long-term care facility patients and discussions among the researchers. After each interview, the two researchers held discussions to review the interview content and atmosphere and to identify any additional questions or issues to be addressed in subsequent interviews.

The first researchers and participants wore N95 masks to avoid infection transmission during the interviews. Interviews were conducted at locations chosen by the participants in comfortable, well-ventilated spaces. Interviews were conducted outdoors or in locations where windows could be opened for ventilation. One interview took place in the participant’s home, five were held in cafés near the participants’ homes or workplaces, and three were conducted in the office of a participant or acquaintance. Before beginning each interview, a researcher explained to the participant that the entire interview would be recorded and that the recorded data would be transcribed verbatim. Consent was obtained from all participants. Open-ended questions were used to encourage the participants to describe their experiences in their own words. To create a warm atmosphere at the beginning of each interview, the researchers introduced themselves and engaged in casual conversations. The interview began with the open-ended question, “As a caregiver who admitted a family member to a long-term care hospital, can you talk freely about your experiences during the COVID-19 pandemic?” During the interviews, the researchers minimized artificial intervention, allowing the participants to speak freely about their experiences. Nonverbal communication and brief responses were used to encourage participants to continue their stories. Supplementary questions were asked when participants had difficulty expressing themselves or when additional details were needed. Supplementary questions included the following: “What was your visitation experience like before and after the COVID-19 pandemic?” “How did the altered visitation guidelines during the COVID-19 pandemic affect you?” “What measures did you take when visiting was not possible because of the pandemic?”.

The interviews concluded following the ninth interview due to a repetition in responses. At this point, the researchers determined that data saturation had been reached. The interviews lasted 50–94 min, averaging 69 min. All interviews were recorded using two devices, and the content was transcribed verbatim on the same day. We also recorded nonverbal expressions such as facial expressions, distinctive gestures, and the overall interview atmosphere in the memo notes, which were used for reference during the analysis. All personal information was anonymized to protect the participants’ identities, and unique identification codes were assigned before being stored in computer files.

### 2.4. Ethical Considerations

This study was approved by the Institutional Review Board of Changwon National University (7001066-202103-HR-011) on 19 April 2021 to ensure compliance with the standard protocol for studies involving human participants. Before data collection, the purpose, methods, and procedures of the study were explained to the participants to guarantee their voluntary participation. We informed participants that their privacy and anonymity would be protected and that they could withdraw from the interview at any time. The participants were also informed that the collected data would be stored securely and destroyed in accordance with IRB regulations after the conclusion of the study, and written consent was obtained from all participants.

### 2.5. Data Analysis

We analyzed the collected data using Colaizzi’s phenomenological method [22]. The researchers immersed themselves in the data by reading the interview transcripts multiple times to gain a thorough understanding of the participants’ experiences. During transcription, key phrases or sentences related to the phenomenon were identified and extracted as significant statements for further analysis. These significant statements were then transformed into formulated meanings through discussions among the researchers, in order to interpret the participants’ words and uncover underlying meanings. The formulated meanings were grouped into themes, reflecting the different perspectives of the phenomenon, and then clustered to represent the participants’ shared experiences. To ensure the validity of the structure, two participants were asked to provide feedback in order to confirm that the analyzed content accurately reflected their experiences.

### 2.6. Ensuring Rigor and Researcher Preparation

The evaluation criteria proposed by Lincoln and Guba [27] were applied to ensure the rigor of this qualitative study. We aimed to ensure credibility, transferability, auditability, and confirmability throughout the interview and analytical processes. During the research process, the researchers maintained a neutral stance by bracketing their clinical work experience to prevent them from influencing the results. To ensure reliability, participants were debriefed at the end of each interview to confirm the content, and member checks were conducted with two participants to verify the analysis results. To ensure transferability, we recruited participants from various professional backgrounds, age groups, genders, and relationships with patients, thereby enhancing the generalizability of the study’s findings. Furthermore, we presented the results to the primary caregiver of a long-term care hospital patient who had not participated in the research and asked them to verify whether the findings were meaningful and applicable to their experiences. To confirm applicability, the analysis process and findings were reviewed by a nurse with extensive experience as a head nurse in a long-term care hospital and expertise in qualitative research. Auditability was achieved by requesting an evaluation of the research results from a nursing professor with extensive experience in phenomenological analysis. For confirmation, the researchers held ongoing discussions to exchange opinions about the research. The researchers worked to maintain objectivity by limiting their literature review to identifying research trends and avoiding preconceived notions that could have influenced the results.

The first researcher is a registered nurse with seven years of clinical experience and three years of experience as an epidemiological investigator. During COVID-19, she received nursing education and responded to outbreaks in vulnerable facilities, including long-term care hospitals. The first researcher also gained foundational knowledge about phenomenological research methods by taking graduate-level qualitative research methodology courses and participating in qualitative research seminars and workshops. The second researcher has taught graduate-level qualitative research methods and provided nursing education during the COVID-19 pandemic. The second researcher has conducted numerous qualitative studies, particularly on nursing phenomena during COVID-19, and has published research findings in peer-reviewed academic journals. Both researchers were well prepared for research in this area. In this study, disagreements were resolved through the continuous discussion and exchange of opinions between the two researchers to maintain rigor. The researchers tried to omit bias from their clinical experience and carried out discussions to minimize preconceptions that could affect the study results. Through this process, the researchers’ differences in interpretation were reconciled, and a consistent understanding of the data analysis was achieved.

## 3. Results

We identified 214 significant statements, and similar statements were grouped into themes. Three comprehensive themes and eight theme clusters were identified (Table 3). The final themes were as follows: “a prison-like long-term care hospital bound by strict COVID-19 prevention rules”; “growing affection for unreachable parents”; and “adapting to a new, safer daily life.”

### 3.1. Theme 1: A Prison-like Long-Term Care Hospital Bound by Strict COVID-19 Prevention Rules

This theme pertains to the participants’ perceptions of hospital visitation policies during the early stages of the COVID-19 pandemic when they chose to keep their parents in long-term care hospitals. Initially, they were not allowed to visit in person. When visits resumed, stringent rules and restrictions left them unable to touch their loved ones, who were separated by transparent glass, leading to a feeling akin to being in prison. Nevertheless, they comforted themselves by believing this was the best way to protect their vulnerable parents.

#### 3.1.1. Difficult Visitation Processes and Limited Visiting Hours

Participants found that, during the initial outbreak of COVID-19, they were unable to visit their parents in person. Later, when visits became possible again, strict visitation rules required them to contact the hospital in advance to schedule a noncontact visit during preset times and days when no other patients had reservations. This led to a significant decrease in the number of visits compared with before the pandemic, leaving caregivers frustrated that they could not see their loved ones more frequently. The restricted visitation times added further stress, as caregivers often felt rushed and uncomfortable after brief encounters.


*“[The noncontact visitation] is only 10 min. Just 10 min. I can’t touch, and I was constantly watching the clock. After 10 min, I have to leave because others are waiting. So, even while looking at my mother, I feel anxious. I should ask more and see her face more. But my mother, she just cries for 10 min, and then it’s over. She cries…it’s very upsetting. Even after leaving, I feel uneasy”.*
(P8)


*“I can’t visit in person, so all the visits are noncontact. And if I want to arrange a noncontact visit, I have to book it days in advance”.*
(P3)

#### 3.1.2. Unable to Touch Parents and Blocked by a Glass Partition

Even after successfully scheduling visits, the participants could not physically touch their parents because of the glass partition, which acted as a barrier. This hindered smooth communication and caused frustration. The experience of seeing their parents through the glass left caregivers feeling even more distant, and many expressed fear of the next visit, as their parents often grew more anxious about being discharged from the nursing home. Seeing their parents confined behind the glass gave them a sense of separation, similar to families divided by a border, making the experience emotionally painful.


*“Now, since direct meetings between caregivers and patients aren’t allowed, all I can do is go and see [my mother], but it feels awkward. I really need to see and hold her hand after the meeting. But there’s this glass partition, and if not that, then we have to talk over the phone. It’s not a school or a prison. It feels more uncomfortable…. Honestly, it’s better not to see her at all. When I do, I feel uneasy for the rest of the day, and I can’t focus on work”.*
(P9)


*“After a noncontact visit, the older adult often experiences emotional swings. Sometimes, I feel scared about visiting because I’m worried they won’t adapt”.*
(P3)

#### 3.1.3. A Haven Amid COVID-19

While the participants expressed dissatisfaction with the limitations of noncontact visits in terms of time and space, they also recognized that these measures were necessary to protect their vulnerable parents from COVID-19.


*“I believe the visitation guidelines are necessary. It’s very frustrating and hard, but that’s the only way to protect the patients in nursing homes and keep them healthy. It’s something we just have to accept”.*
(P1)


*“There are so many vulnerable patients in the hospital, so we have to avoid infection. Normal people may not understand, but it’s the right thing to do for everyone’s safety”.*
(P9)

### 3.2. Theme 2: Growing Affection for Unreachable Parents

This theme captures changes in the participants’ perceptions of and emotional responses toward their hospitalized parents. The inability to enter the hospital and see day-to-day life inside, along with the gradual erasure of their parents from their daily routines, made the participants feel increasingly distanced. Participants, who were growing weary due to the prolonged pandemic, also experienced guilt as they watched their parents struggle with the closed-off environment of the long-term care hospital.

#### 3.2.1. Having to Entrust Parents to Medical Staff and Caregivers Fully

Before the pandemic, participants could observe their parents’ day-to-day lives in the hospital. With no way to see inside, they often felt uneasy about the caregivers’ attitudes. They had no choice but to rely on medical staff and caregivers for updates about their parents’ condition and treatment, leaving them feeling that their parents’ lives in the nursing home were hidden in a fog.


*“Before, I could go in and directly check on my mother’s care, which gave me peace of mind since I could see for myself. Now, I just talk to her through the glass, and I feel a bit anxious, not knowing how she’s really doing”.*
(P2)


*“I don’t know what the caregivers are doing. I just have to trust them since I can’t go in and check for myself”.*
(P6)

#### 3.2.2. The Fading Presence of Parents in Daily Life

As the pandemic dragged on, participants became mentally exhausted, and their parents’ presence began to fade from their daily lives in an “out of sight, out of mind” manner.


*“If I were seeing her, I’d feel emotions like guilt, but since I can’t, I’ve just become indifferent. Honestly, I think I’ve grown numb. Since I can’t see her, it’s like she’s gradually slipping away”.*
(P5)

#### 3.2.3. Meaningless Screams in Confinement

Unable to see their families as often as before, owing to the strict regulations, and confined to their rooms without the freedom to move about the hospital or take walks, parents often complained about feeling trapped and begged for discharge. However, caregivers had no choice but to deny these requests, knowing they could not care for their parents. This left them heartbroken.


*“During the first year of the pandemic, my mom kept asking to go home, saying she wanted to be discharged and live at home again. She said it every day, and my brother and I took turns calling her, but it was so painful to hear”.*
(P6)


*“When I heard that the nurses were confining the patients, even running over if the door opened, I felt guilty. I felt like I was putting my parents through unnecessary suffering by keeping them there”.*
(P3)

Participants also felt bitter when their parents began to doubt their sincerity, suspecting that their children were using COVID-19 to avoid visiting.


*“Not being able to see us is the hardest part for them. They think we’re making excuses, that we’re lying about not being able to visit because of COVID. They start to distrust us…. It feels like my efforts aren’t being understood. When they say we’re using COVID as an excuse, it really hurts”.*
(P3)

### 3.3. Theme 3: Adapting to a New and Safer Daily Life

This theme focuses on how the caregivers have adjusted to prolonged pandemics. Their sense of guilt over past actions deepened their affection for their parents, while the availability of vaccines allowed them to hope for a return to everyday life. Caregivers also found new ways to stay in touch with their loved ones and created new routines during the pandemic.

#### 3.3.1. Strengthening Familial Bonds Despite Fewer Visits

Participants reported feeling more affectionate toward their parents and regretting what they had not performed before their parents were admitted to a nursing home. Despite being unable to visit as often, they found that their family bonds grew stronger, and they reached out to their parents more frequently to check on them.


*“I can’t visit as often, so I feel even sorrier for her now”.*
(P2)


*“I still regret not doing more for them when they were healthy, like buying supplements or taking them on a trip. I feel sorry about it all the time”.*
(P3)


*“Since I can’t visit often, I worry about her more, and I feel more affectionate toward her”.*
(P1)

#### 3.3.2. Creating and Anticipating New Routines

Participants began to find ways to stay connected with their parents during the pandemic. Since visits were limited, some video calls were arranged using the hospital’s phone, or photos were received from the caregivers. Others purchased cell phones to maintain contact with their parents, discovering new ways to ease their anxiety. They also began preparing for potential long-term crises by exploring ways to maintain communication with their loved ones.


*“We got my mom a smartphone for video calls, but the staff didn’t always have time to help, so I bought her a phone so we could stay in touch directly”.*
(P8)


*“Just in case this goes on longer, I’ve already set up another phone for my mom so we can have video calls in the future”.*
(P3)

With the rollout of the COVID-19 vaccine, participants began to hope for a return to normalcy, where they could visit their parents more freely.


*“My mom’s vaccinated, so I’m hoping things will loosen up, and maybe if the staff and patients all get their second doses, visitation rules will relax, too”.*
(P2)


*“Once we’re all vaccinated, things will get better. We’ll be able to meet again, hold hands, and eat together. Things will improve soon”.*
(P4)

## 4. Discussion

Through phenomenological analysis, this study revealed the essence of the experiences of the family caregivers of long-term care hospital patients during the early stages of the COVID-19 pandemic. Three main themes emerged that were interconnected in terms of time and concepts.

The first theme, “a prison-like long-term care hospital bound by strict COVID-19 prevention rules”, describes the frustrations caused by the limitations on visitation imposed by the pandemic, including scheduled visits, time and frequency restrictions, and sharp decreases in visitation opportunities. During the early days of the COVID-19 pandemic, South Korea’s nursing home visitation policy was strictly restricted to prevent the spread of infection. Patients were encouraged to communicate with their families primarily through non-face-to-face means, with limited face-to-face visitation permitted only when necessary. Strict conditions such as wearing protective clothing, having their temperature taken, and following quarantine procedures were required. These policies were implemented as essential measures to protect nursing home patients, who were particularly vulnerable to infection [28]. These limitations made it difficult for caregivers to see their loved ones as frequently, leading to emotional strain. Noncontact visits, hindered by glass barriers, added to the distress, as physical touch and smooth communication were not possible. While long-term care hospitals were seen as places that protected residents from infection, caregivers felt that their loved ones were trapped in a prison. Prior research [29] has shown that older adults in institutional care experience physical, social, and psychological changes owing to restricted spaces and controlled routines, especially in long-term care hospitals with multiple residents in a room, strict rules, and communal living arrangements, leading them to feel like prisoners. While infection prevention measures were seen as necessary for protection, long-term care hospitals were also perceived as “safe prisons.” This is consistent with a previous study [24] that noted how infection control measures, while essential, made facilities feel like protective barriers. To address these negative perceptions and emotions, it is important to explore the caregivers’ perspectives in depth, develop comprehensive visitation guidelines, and improve the design of visitation spaces. Additionally, effort should be made to establish systematic responses to both direct and indirect visits through tools such as posters, brochures, and videos.

In the second theme, “growing affection for unreachable parents”, the caregivers’ feelings of guilt, worry, and emotional distance became more pronounced as the pandemic continued. Caregivers became increasingly reliant on the staff of the long-term care hospitals and grew more anxious about the safety of their parents, especially given the vulnerability of the older adult population. As the pandemic progressed, caregivers experienced emotional fatigue, aligning with a previous study [30] which indicated that prolonged illness negatively affects family functioning. Caregivers felt guilty because they could not fulfill their parents’ requests for discharge, and they were troubled by the sight of their parents experiencing the closed-off environment of long-term care hospitals. These feelings of guilt were exacerbated by reduced visitation opportunities, in line with the traditional belief in Korean society that adult children are responsible for providing emotional, instrumental, and financial support to their older adult parents [31]. Family caregivers of long-term care hospital patients often experience internal conflict, guilt, and a sense of burden [32] and, before the pandemic, regular visits were a way to mitigate these emotions. However, COVID-19 forced caregivers to rely on noncontact visits or reduced in-person interactions [33], which intensified their feelings of affection toward their parents. In addition, the pandemic created economic, psychological, and social difficulties for many people around the world, and the financial burden, social isolation, and fear of infection that occurred during the pandemic period have since become even greater burdens for caregivers of patients hospitalized in long-term care hospitals [34]. Caregivers’ increasing dependence on the staff of long-term care hospitals during the pandemic highlights the need for nursing homes to strengthen their caregiving capabilities to ensure that families trust the care provided. Furthermore, providing emotional support and counseling for caregivers, particularly for those who feel guilt and loss, could help alleviate these negative emotions [35,36,37]. Programs focused on caregiver support and education for nursing home staff are necessary, as are strategies to increase opportunities for communication, such as video calls, which allow residents and their families to maintain emotional connections and share everyday life.

The third theme, “adapting to a new, safer daily life”, reflects the caregivers’ gradual acceptance of the new reality, characterized by a mix of affection, regret, and growing hope owing to vaccines. Videos and phone calls, along with photos, allowed caregivers to communicate indirectly with their loved ones, leading to satisfaction. This finding is consistent with the previous studies, showing that video calls improved the life satisfaction, self-integration, and mood of the nursing home residents during the pandemic [29]. Additionally, video calls allow for family-centered care by enabling multiple family members to participate without time or space limitations. Caregivers expressed positive expectations and hope as vaccines became available and noncontact visitation methods diversified. This is consistent with the prior research [38] which showed that caregivers who look for the positive aspects of difficult situations are better equipped to cope with them. Primary caregivers who receive public social services or help from other family members have also been found to have more positive experiences [39], underscoring the importance of psychological support, societal interest, and institutional support.

This study’s exploration of the experiences of the family caregivers of patients in long-term care hospitals during a pandemic holds significance for nursing research. The findings can serve as foundational data for preparing physical facilities to facilitate meetings between the patients and caregivers during future infectious disease outbreaks, developing strategies for noncontact visits, ensuring the quality of services in long-term care hospitals, and seeking interventions to promote the caregivers’ emotional and social well-being. Furthermore, this study provides valuable evidence for the establishment of infection control strategies in long-term care hospitals.

## 5. Conclusions

Nursing care encompasses not only the patients but also individuals, families, and communities. This study delved into family caregivers’ experiences during the COVID-19 pandemic and explored the meanings and implications. Although the COVID-19 pandemic has officially ended, the increasing rates of long-term care hospital admissions in the aging society, as well as the possibility of future infectious diseases, underscore the need to develop robust infection control measures and visitation protocols for long-term care hospitals. Based on our findings, we make the following recommendations to ensure high-quality care for both the patients and their caregivers:

First, in-service and continuing education on human rights and communication must be provided for healthcare workers and other staff in long-term care hospitals. Second, practical guidelines should be established for noncontact visits, and the physical facilities for such visits should be improved. Third, psychological support, institutional support, and nursing interventions must be offered to patients and their caregivers in long-term care hospitals.

A limitation of this study is that its focus was limited to experiences during the early stages of COVID-19; thus, further research is necessary to explore experiences during the later stages or recovery phases related to additional emotional changes, as well as the long-term impacts as the pandemic prolongs, and the development of methods for non-contact visits. Further research is also needed on the caregivers’ perceptions of long-term care hospitals and the infection control systems in place.

Although this study was conducted in the context of the COVID-19 pandemic, its results can be applied to various situations. For example, when strict visiting rules must be followed to ensure safety during the spread of other infectious diseases such as influenza, or when caregivers are unable to visit the hospital in-person due to living far away, the non-face-to-face communication infrastructure can be utilized. Additionally, programs can be introduced to provide caregivers with psychological and emotional support in situations where face-to-face visits are difficult. Educating healthcare workers is also an essential element in the daily operation of hospitals and patient care; hence, such education can continue to be applied in situations other than pandemics. Therefore, the results of this study are expected to be useful in various situation and not only limited to pandemics.

## Figures and Tables

**Figure 1 healthcare-12-02254-f001:**
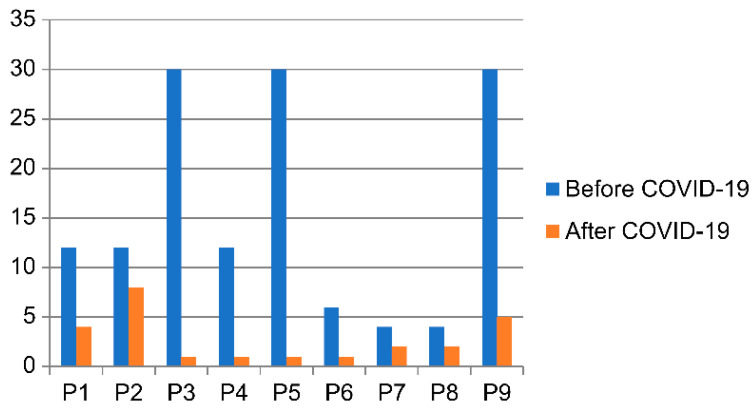
Number of meetings (per month).

**Table 1 healthcare-12-02254-t001:** Criteria for selecting research participants.

Inclusion Criteria	Exclusion Criteria
Caregivers of patients admitted to long-term care hospitals for at least six months at the time of the interview.	Non-immediate family caregivers
Only immediate family members of the patient were eligible to participate.	Caregivers of patients admitted to long-term care hospitals for less than six months were excluded.

**Table 2 healthcare-12-02254-t002:** General Characteristics of Participants (N = 9).

Characteristics of Participants (Caregiver)						Characteristics of Patients		Number of Meetings	
								(Per Month)	
No	Sex	Age	Education	Occupation	Relationship	Sex	Hospitalization	Before	After
							Period	COVID-19	COVID-19
							(Year)		
1	Female	51	Bachelor’s	Employee	Daughter	Female	12	12	4
2	Female	48	Bachelor’s	Homemaker	Daughter	Female	2	12	8
3	Female	54	High-school	Homemaker	Daughter	Male	6	30	1
						Female			
4	Female	62	High-school	Employee	Daughter-in-law	Female	7	12	1
5	Male	55	Bachelor’s	Employee	Son	Female	5	30	1
6	Female	62	High-school	Homemaker	Daughter	Female	9	6	1
7	Female	53	High-school	Homemaker	Daughter	Female	6	4	2
8	Female	54	Bachelor’s	Homemaker	Daughter-in-law	Female	3	4	2
9	Male	61	High-school	Employee	Son	Female	7	30	5

**Table 3 healthcare-12-02254-t003:** Experience of inpatient caregivers in long-term care hospitals during the COVID-19 pandemic.

Categories	Theme Clusters
Prison-like long-term care hospital bound by strict COVID-19 prevention rules	Difficult visitation processes and limited visiting hours
	Unable to touch parents, blocked by a glass partition
	A safe haven amidst COVID-19
Growing affection for unreachable parents	Having to fully entrust parents to medical staff and caregivers
	The fading presence of parents in daily life
	Meaningless screams in confinement
Adapting to a new and safer daily life	Strengthening familial bonds despite fewer visits
	Creating and anticipating new routines

## Data Availability

Data sharing is not applicable to this article due to the small number of participants involved. The study’s design prioritizes the protection of participant identities, therefore, individual data will not be shared.

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
