# Peer review of "Experience of Family Caregivers in Long-Term Care Hospitals During the Early Stages of COVID-19: A Phenomenological Analysis"

_healthcare, 2024, doi:10.3390/healthcare12222254_

Round 1
Reviewer 1 Report
Comments and Suggestions for Authors
Specific Comments:
Abstract:
1. Abstract is well written but its conclusion mentions that the findings will support communication during outbreaks, it looks like somewhat disconnected from the main themes of emotional stress and family separation. Authors should specify how these findings can be applied.
Introduction:
2. The introduction provides general overview of COVID-19, but it needs to focus more on the relevance of this pandemic to long-term care hospitals. Rather than general statistics about depression and anxiety in South Korea, authors should provide data specifically related to long-term care facility caregivers and patients.
3. Authors should add clear statement of research problem. What are the specific gaps in current research regarding caregivers’ experiences during the COVID-19 pandemic?
Methods:
4. Authors should provide the inclusion and exclusion criteria. Authors may add sample size calculation with justification.
5. It would be helpful to include a stronger rationale for why this phenomenological method (Colaizzi’s method) was chosen. The use of snowball sampling may introduce bias by limiting the diversity of experiences represented in the study. Authors should justify how participant bias was minimized.
Results:
6. Results are poorly described. Authors should add figures and tables to clearly present their findings. The results should be presented after robust statistical analyses.
Discussion:
7. The discussion should include a more detailed analyses. For example, could the observed emotional responses be attributed to factors other than visitation restrictions, such as the caregivers' own health or external stressors during the pandemic?
8. Authors should discuss the limitations of the study.
General Comments:
1. Ensure proper sentence structure with valid grammar.
2. Incorporate recent and updated references.
3. Add abbreviations list.
Comments on the Quality of English LanguageEnsure proper sentence structure with valid grammar.
Reviewer 2 Report
Comments and Suggestions for Authors
Very interesting article.
It explores a very pertinent topic and uses the phenomenological approach, which allows us to explore a subject in depth.
Simple but with a wealth of content, addressing a subject that has had such an impact on family members and users themselves during the covid-19 pandemic.
Title - clear and objective
Abstract - objective, uses clear language and explains the type of work carried out and the main conclusions.
Key words - I don't think it makes sense to include South Korea as neither the title nor the abstract identifies the place where the study was carried out. I would also suggest putting COVID-19 Pandemic all together or choosing just one word. If you want to be more objective, you could put phenomenology instead of qualitative research.
Theoretical development well, concluding with the defined research question and objectives.
They clearly define the inclusion criteria for participants.
Address
They refer to how the interviews were carried out, the selection of the location and the care taken to minimise the risk of infection.
They refer to ethical issues: informed consent and anonymity.
Well-defined data collection period.
They mention the number of interviews and the average duration.
They give a brief overview of the content analysis method used - Colaizzi's method.
They took care to validate the information gathered with two of the participants.
They identify eight themes and three categories, and in the discussion seek to explore them with the contribution of other authors.
A clear and objective conclusion.
Bibliography: they should standardise the bold in the dates; of the 33 references, they present 9 references with a date greater than or equal to 2020. They include 3 very old references - 1978, 1985 and 1987. According to the text in which they are referenced, I don't think it's justified to put them in, since they are never cited individually, and you should opt for the most up-to-date reference. The reference to Colaizzi's method is old but may be the author's original book, however there may be more current articles on this method.
I think there will be more up-to-date bibliography on this subject and they must cahnge some of them.
Reviewer 3 Report
Comments and Suggestions for Authors
This manuscript explored the lived experiences of family caregivers of patients residing in long-term care hospitals in South Korea during the COVID-19 pandemic's early stages. Employing a phenomenological approach, the researchers conducted in-depth interviews, ultimately identifying three core themes: the hospital as a restrictive environment, evolving emotional responses toward isolated parents, and adaptation to a new reality.
1. Main Question and Clarity:
The central research question is somewhat implicit: How did the COVID-19 pandemic and its associated restrictions impact the experiences of family caregivers with loved ones in long-term care hospitals? While not explicitly stated as a question, the manuscript adequately addresses this throughout.
2. Originality and Relevance:
The manuscript offered valuable insights into an under-researched area: the unique challenges faced by family caregivers of long-term care patients during the pandemic's early stages. The focus on South Korea adds geographical specificity and potential for comparison with other contexts. While existing literature acknowledges caregiver burdens broadly, this work highlights how pandemic-related restrictions exacerbated these, contributing to feelings of isolation, helplessness, and emotional fatigue.
3. Missing Literature:
The manuscript would benefit from incorporating additional studies to contextualize its findings further. For instance:
- International perspectives on caregiver experiences during COVID-19: Research such as "The impact of COVID-19 on family caregivers of people living with dementia: An international qualitative study" (C. Brown et al., 2021) could provide valuable comparisons.
- Studies exploring interventions to mitigate caregiver burden: Inclusion of work like "Effectiveness of psychosocial interventions for informal caregivers of older adults during the COVID-19 pandemic" (A. Lloyd et al., 2022) could strengthen the discussion on practical implications.
4. Methodology and Design:
- Sampling methodology: While snowball sampling can be helpful in qualitative studies, it might introduce bias. The researchers acknowledged using this but should clarify how they mitigated potential limitations (Lines 96-98).
- Sample size: The sample size of nine participants is small, potentially limiting the generalizability of the findings. While data saturation is mentioned (Line 134), including a larger, more diverse sample would enhance the study's robustness. Any take on this?
- Data analysis: The authors adequately described applying Colaizzi's method but could elaborate on how they ensured intercoder reliability during the coding and theming process.
5. Conclusions and Evidence:
The study's conclusions are generally well-supported by the themes identified through the interviews. However, the connection between themes could be further strengthened.
- For example, the second theme, "growing affection for unreachable parents," could benefit from exploring the complex interplay between increased affection, feelings of guilt, and helplessness due to forced separation (Lines 249-255).
- The authors appropriately mentioned that caregivers expressed hope with the advent of vaccines (Lines 318-323). However, this finding needs more nuance, recognizing that the pandemic's prolonged nature continued to impact caregiver well-being even after vaccine availability.
6. Tables and Figures:
The manuscript did not include any figures. The inclusion of a visual representation of the themes and their interrelations could enhance the clarity and impact of the findings.
7. General Comments and Weaknesses:
- Limited temporal scope: The study focuses on the early stages of the pandemic. Revisiting these experiences now, with a longer-term perspective on how the pandemic reshaped caregiver realities, would be beneficial.
- Lack of demographic data: While the manuscript mentions participant characteristics briefly (Lines 99-101), providing more details on factors such as caregivers' age, gender, and socioeconomic background would enable richer analysis.
- Overemphasis on caregiver burden: While highlighting challenges is essential, the manuscript would benefit from incorporating positive aspects of caregiving, resilience strategies employed by caregivers, and lessons learned during this challenging period.
Specific Comments (Line Numbers):
- Line 33: The statement that "mask-wearing is still recommended... for vulnerable populations" requires updating and contextualizing to current public health recommendations in South Korea.
- Line 76: Clearly distinguish the terms "long-term care facility" and "long-term care hospital," as their functions and regulatory contexts differ.
- Lines 198-205: Elaborate on the specific visitation policies in South Korea that were implemented in long-term care hospitals. This information is crucial for understanding the context of caregivers' experiences.
Reviewer 4 Report
Comments and Suggestions for Authors
I have attached some comments. The section describing the data analysis is sparse, and suggest addressing the low number for saturation with a citation. See comments.

Round 2
Reviewer 1 Report
Comments and Suggestions for Authors
Authors have diligently addressed almost all the comments and concerns raised during the review process except the sample size justification. The revisions made have significantly improved the quality of the article. However, there are still need to improve the methodology section. It would be more appropriate to add inclusion/exclusion criteria based upon multiple factors. I recommend to add graphical presentation of inclusion & Exclusion criteria.
Reviewer 3 Report
Comments and Suggestions for Authors
Glad with changes
Author Response
Thank you for your valuable feedback